# Prevalence and Correlates of Pre-Treatment HIV Drug Resistance among HIV-Infected Children in Ethiopia

**DOI:** 10.3390/v11090877

**Published:** 2019-09-19

**Authors:** Birkneh Tilahun Tadesse, Olivia Tsai, Adugna Chala, Tolossa Eticha Chaka, Temesgen Eromo, Hope R. Lapointe, Bemuluyigza Baraki, Aniqa Shahid, Sintayehu Tadesse, Eyasu Makonnen, Zabrina L. Brumme, Eleni Aklillu, Chanson J. Brumme

**Affiliations:** 1Department of Pediatrics, College of Medicine and Health Sciences, Hawassa University, Hawassa 1560, Ethiopia; 2Division of Clinical Pharmacology, Department of Laboratory Medicine, Karolinska Institute, Karolinska University Hospital Huddinge, 141 86 Stockholm, Sweden; 3Faculty of Health Sciences, Simon Fraser University, Burnaby, BC V5A 1S6, Canada; 4Department of Pharmacology, College of Health Sciences, Addis Ababa University, Addis Ababa 9086, Ethiopia; 5Adama General Hospital and Medical College, Adama 84, Ethiopia; 6Department of Microbiology, SNNPR Reference Laboratory, Hawassa 149, Ethiopia; 7British Columbia Centre for Excellence in HIV/AIDS, Vancouver, BC V6Z 1Y6, Canada; 8Center for Innovative Drug Development and Therapeutic Trials for Africa (CDT Africa), College of Health Sciences, Addis Ababa University, Addis Ababa 9086, Ethiopia; 9Division of Infectious Diseases, Department of Medicine, University of British Columbia, Vancouver, BC V5Z 1M9, Canada

**Keywords:** HIV, pediatrics, Ethiopia, pre-treatment drug resistance, combination antiretroviral therapy (cART), dried plasma spots, dried blood spots

## Abstract

Pediatric human immunodeficiency virus (HIV) care in resource-limited settings remains a major challenge to achieving global HIV treatment and virologic suppression targets, in part because the administration of combination antiretroviral therapies (cART) is inherently complex in this population and because viral load and drug resistance genotyping are not routinely available in these settings. Children may also be at elevated risk of transmission of drug-resistant HIV as a result of suboptimal antiretroviral administration for prevention of mother-to-child transmission. We investigated the prevalence and the correlates of pretreatment HIV drug resistance (PDR) among HIV-infected, cART-naive children in Ethiopia. We observed an overall PDR rate of 14%, where all cases featured resistance to non-nucleoside reverse transcriptase inhibitors (NNRTIs): ~9% of participants harbored resistance solely to NNRTIs while ~5% harbored resistance to both NNRTIs and nucleoside reverse transcriptase inhibitors (NRTIs). No resistance to protease inhibitors was observed. No sociodemographic or clinical parameters were significantly associated with PDR, though limited statistical power is noted. The relatively high (14%) rate of NNRTI resistance in cART-naive children supports the use of non-NNRTI-based regimens in first-line pediatric treatment in Ethiopia and underscores the urgent need for access to additional antiretroviral classes in resource-limited settings.

## 1. Introduction

Morbidity and mortality associated with human immunodeficiency virus (HIV) infection have substantially decreased with the introduction of effective combination antiretroviral therapy (cART) [1,2,3]. Towards realizing the Joint United Nations Programme on HIV/AIDS (UNAIDS) “90-90-90” targets to help end the acquired immunodeficiency syndrome (AIDS) epidemic [4], which outline ambitious goals for timely HIV diagnosis, sustained treatment, and maintenance of virologic suppression to limit onward viral transmission [5], the number of HIV infected individuals on cART is estimated to have reached 66% globally in 2018 [6]. Ensuring the sustained effectiveness of cART is key to maintaining these gains and is particularly critical in settings where cART options remain relatively limited. 

Pediatric HIV treatment, especially in resource-limited settings, remains a major challenge to achieving and sustaining global treatment and virologic suppression targets [7]. This is in part due to the complexities of administering cART in children combined with the lack of routine availability of viral load and genotypic drug resistance testing to guide and monitor HIV treatment efficacy in resource-limited settings. Moreover, children in such settings may be particularly vulnerable to harboring pretreatment HIV drug resistance mutations, as antiretrovirals prescribed during pregnancy for prevention of mother-to-child transmission (PMTCT), if suboptimally administered, could lead to selection and transmission of HIV drug resistance mutations [8,9].

In Ethiopia, prevalence of HIV infection in the adult population is 1.4%, though there is substantial urban–rural variation [10]. The provision of cART to pregnant women for PMTCT in Ethiopia has been steadily increasing, and coverage reached 50–69% in 2017 [11]. This is partially the result of the 2013 nationwide implementation of PMTCT “option B+”, which recommends initiating lifelong cART during pregnancy and providing nevirapine for six weeks for the neonate with exclusive breast feeding for the first six months and complementary feeding thereafter [12]. Surveillance for pretreatment HIV drug resistance (PDR), also called pretherapy drug resistance [13], is therefore critical to the care and the treatment of HIV-infected infants and children; however, no studies have assessed burden of PDR among HIV infected children in this region.

In Ethiopia, neither routine plasma viral load monitoring nor HIV drug resistance testing are readily available to guide individualized patient care. Empiric choice of cART is therefore the routine practice, and for this reason, understanding the overall burden and the pattern of PDR among HIV-infected children could help inform empiric treatment guidelines and practices. Towards this goal, the current study assessed the prevalence and the correlates of PDR among HIV-infected cART-naive children from a resource-limited setting in Ethiopia using dried plasma and blood samples (DPS and DBS, respectively). 

## 2. Materials and Methods 

### 2.1. Study Participants

The present study comprised children who were originally screened for eligibility for inclusion in the Efavirenz Pediatric Dose Optimization Study (EPDOS) cohort. EPDOS enrolled cART naïve HIV-infected children at seven HIV/AIDS treatment centers across two of the largest administrative regions in Ethiopia: Oromia and Southern Nations Nationalities and Peoples Region (SNNPR), whose combined populations exceed 55 million [14]. The centers in Oromia region were: Adama General Hospital, Asela Referral and Teaching Hospital, and Shashemene General Hospital. From SNNPR, they were: Hawassa University Comprehensive Specialized Hospital, Adare General Hospital, Otona Referral Hospital, and Arbaminch General Hospital. These regions were selected because of the high HIV prevalence and the large population. In Ethiopia, the estimated HIV pediatric population is 62,000 (38,000–86,000) [15]. Children who had tuberculosis and those who had previously been on combination antiretroviral therapy were not eligible to be enrolled in the parent EPDOS study, though children with PMTCT exposure would have been eligible if data were available. Note, however, that PMTCT history was not available for study participants.

Beginning in 2001, Ethiopia adopted PMTCT intervention Option A, under which eligible pregnant women with CD4 < 350 copies/mm^3^ were initiated on cART. At this time, women who did not meet the CD4-based eligibility criteria were provided antepartum zidovudine (AZT) and intrapartum single dose nevirapine. In 2013, the guidelines were amended to recommend Option B+, and uptake has scaled up since then. By 2014, around 2500 health facilities had started providing PMTCT services. Currently, PMTCT coverage estimates in Ethiopia vary between 50–70% depending on region [11,16,17].

A total of 117 children were screened for EPDOS, of whom 111 were eventually enrolled [14]). The present study analyzed the baseline cross-section of a subset of the children originally screened for EPDOS; specifically, 93 (of 117; 79.5%) participants for whom a pre-cART sample was available or could newly be obtained. Participants were between the ages of 3–18 years, were cART naïve, and had no acute severe illnesses at enrollment. Since Ethiopia practices the “test and treat” strategy, participants were enrolled within 2–4 weeks of HIV diagnosis after the required pretreatment counseling was completed. Moreover, we could confirm that all children were indeed cART naive at time of enrollment, as HIV care and treatment in Ethiopia is managed under a centralized system. 

Sociodemographic, clinical, and laboratory data were collected at baseline. Viral loads were determined using the RealTime HIV-1 viral load test (Abbott, Des Plaines, IL, USA).

### 2.2. Ethics Statement

Ethical approval for this study was obtained from the Institutional Ethics Review Boards of Addis Ababa University College of Health Sciences, Karolinska Institutet in Stockholm, Sweden, Simon Fraser University and Providence Health Care/University of British Columbia. The renewed National Research and Ethics Review Committee Ethics certificate is SHE/SM/14.3/0421/1/2019. Blood samples were collected after obtaining written informed consent in accordance with the tenets of the Declaration of Helsinki. For participants ≤ 12 years, written informed consent was obtained from their parent or guardian, while for participants > 12 years of age, consent was obtained from the parent or guardian and assent obtained from the participant. All informed consent documents were provided in the local language.

### 2.3. Specimen Collection, Handling and Storage

Up to 5 blood spots (DBS) for 22 HIV infected children and up to 5 plasma spots (DPS) for 71 HIV infected children were collected. Each spot contained approximately 50 µL. The DBS were prepared from participants by fingerprick on blood spot cards (Labmate, Cape Town, South Africa) and dried overnight at room temperature. The DPS samples were prepared by thawing previously-collected plasma samples stored at −80 °C, spotting ~50 µL aliquots on blood spot cards, and drying overnight at room temperature. Each card was individually packed in a plastic specimen bag with desiccant pack and shipped to Simon Fraser University (SFU) for HIV drug resistance genotyping. Spots were stored at room temperature until shipment to Simon Fraser University (SFU); upon receipt, they were stored at −80 °C until tested.

### 2.4. HIV Drug Resistance Genotyping and Phylogenetic Inference

Using a standard ¼” manual hole punch or a pair of metal forceps, two spots (plasma or blood, as provided) per participant were transferred into sterile tubes for nucleic acid extraction [18]. The hole punch was cleaned of residual material between participant cards by punching 10 holes into clean filter paper [18]; the forceps were cleaned using bleach. Total nucleic acids were extracted using the NucliSENS easyMAG System according to manufacturer’s instructions (BioMerieux, Marcy-l’Étoile, France). 

HIV Protease and a portion of reverse transcriptase (RT) spanning a minimum of codons 1–234 were amplified using an initial reverse transcriptase step (Expand Reverse Transcriptase; Roche, Basel, Switzerland) followed by nested PCR (Expand Hifi System; Roche) or alternatively by RT-PCR (using the SuperScript III One-Step RT-PCR System with Platinum Taq High Fidelity DNA Polymerase; Invitrogen, Massachusetts, USA) followed by nested PCR (using the Expand HiFi System; Roche) [19]. Amplification was attempted using up to 4 oligonucleotide primer sets designed to amplify various HIV-1 group M subtypes (Table 1). If amplification failed using the primary set, amplification was attempted using the backup sets. If amplification failed again, fresh nucleic acid extracts were prepared from remaining DPS or DBS, and amplification was re-attempted as above. Amplicons were visualized on a 1% agarose gel and bulk (directly) sequenced on a 3130xl or 3730xl automated DNA sequencer (Applied Biosystems, Foster City, CA, USA). Chromatograms were analyzed using Sequencher version 5.0.1 (Gene Codes, Ann Arbor, MI, USA) or the automated basecalling software RECall [20], where nucleotide mixtures were called if the secondary peak exceeded 25% of the dominant peak height (Sequencher) or 17.5% of the dominant peak area (RECall).

HIV sequences were aligned using HIV Align (options: MAFFT, codon-alignment) [21] and manually inspected using AliView [22]. Maximum likelihood phylogenies were inferred from HIV sequence alignments using PhyML under a general time reversible model of nucleotide substitution [23]. Phylogenies were generated from full alignments as well as alignments stripped of all codons associated with HIV surveillance drug resistance mutations [24] to control for any potential effects of these mutations on tree topology [25]. Phylogenies were visualized using Figtree (version 1.4.4). HIV subtype determination was performed using the Recombinant Identification Program tool hosted by the Los Alamos HIV Sequence Database (LANL) [26]. HIV sequences were deposited into GenBank (accession numbers MN244083–MN244139).

### 2.5. Drug Resistance Genotype Interpretation

Drug resistance genotype interpretation was performed using the Calibrated Population Resistance (CPR) tool on the Stanford University HIV Drug Resistance Database [27,28,29], which identifies mutations conferring resistance to protease inhibitors (PIs), nucleoside reverse transcriptase inhibitors (NRTIs), and non-nucleoside reverse transcriptase inhibitors (NNRTIs) as defined by the World Health Organization (WHO) 2009 list of mutations for surveillance of pretreatment HIV drug resistance [24]. 

### 2.6. Statistical Analysis

Statistical analyses were conducted using GraphPad Prism (version 8, San Diego, CA, USA). The dependent variable was PDR (yes/no) for NRTI, NNRTI. or any resistance. Associations between PDR and sociodemographic, clinical, and laboratory variables were assessed using Fisher’s exact test for categorical independent variables and the Wilcoxon Rank Sum test for continuous independent variables. *p*-values < 0.05 were considered statistically significant.

## 3. Results

### 3.1. Patient Characteristics

A total of 93 children were included in the study; their characteristics at enrollment and the number of participants for which data were available for each characteristic are summarized in Table 2. Their median age was 9 years [interquartile range (IQR): 5–12]; 48 (51.6%) were male. Upon clinical evaluation, 56/85 (65.9%) were identified as having symptoms that define WHO clinical stage 2 or above (Table 2). These symptoms included papular pruritic eruptions (21/85; 24.7%), mucocutaneous viral infections (24/85; 28.2%), chronic diarrhea (18/83; 21.2%), and features of fungal infection (15/85; 17.7%). The median CD4+ T-lymphocyte cell (T-cell) count was 319 (IQR: 141–615) cells per mm^3^, and the median viral load was 4.3 log_10_ copies/mL (IQR: 3.7–4.9 log_10_ copies/mL) in the 41 and 83 children where CD4 and pVL data were available, respectively. There were two cases with hepatitis C virus (HCV) coinfection, but no hepatitis B virus (HBV) coinfection was reported. Regarding nutritional status at HIV diagnosis, 30/82 (36.6%), 23/82 (28.1%), and 20/53 (37.7%) participants were stunted, wasted, and underweight, defined as Z score < −2, respectively.

### 3.2. Prevalence of PDR and Detected Drug Resistance Mutation Types 

Dried blood spots (DBS; *N* = 22) or dried plasma spots (DPS; *N* = 71) were collected from all 93 study participants. HIV drug resistance genotyping was successful for 57/93 (61.3%) of these samples, an overall success rate that is comparable to other studies utilizing dried blood products as starting material [30,31,32]. Success rates did not differ between DPS (14 of 22; 63.6%) and DBS (43 of 71; 60.6%) (Fisher’s exact test, *p* = 1). Somewhat surprisingly, samples for which resistance genotyping was successful did not have significantly different viral loads from those samples where resistance was not successful [median 4.3 (IQR: 3.8–5.0) vs. 4.4 (IQR: 3.2–4.9) *p* = 0.47]. Moreover, genotyping success rate was not significantly associated with any of the participant characteristics listed in Table 2, with the exception of weight-for-age Z-score. Participants for whom genotyping was successful exhibited a median Z-score of −1.9 [IQR −2.9–(−0.9) compared to −1 (IQR −1.7–(−0.1)] for whom genotyping was unsuccessful (Mann–Whitney U-test *p* = 0.03), though we acknowledge that this may be a chance finding. Despite using dried blood or plasma as starting material, some sequences nevertheless bore evidence of amplification of within-host sequence diversity. The number of nucleotide mixtures detected in successful genotypes ranged from 0 to 40 (median 0, IQR 0–14). One of the isolated sequences (EPDOS_9) was defective with an internal stop codon near the end of the sequence.

Consistent with the HIV epidemic in Ethiopia being predominately composed of subtype C [33,34,35,36,37,38], a total of 54 of 57 (94.7%) participants harbored HIV subtype C, one (1.7%) harbored an AG recombinant, one (1.7%) harbored an ACG recombinant, and one (1.7%) harbored subtype A (Figure 1). With the exception of one sibling pair whose viral sequences clustered closely together on the phylogeny, no other phylogenetically-linked infections were observed, as would be expected in a pediatric cohort. Tree topology was not substantially impacted by the presence of drug resistance codons).

Overall, 8/57 successfully genotyped participants harbored HIV drug resistance mutations, yielding a total PDR prevalence of 14% (95% CI: 4.8–24.0%) in our study (Figure 1). All eight cases of PDR featured resistance to NNRTIs; specifically, five (62.5%) solely harbored (NNRTI) resistance mutations while three (37.5%) harbored both NRTI and NNRTI resistance mutations. No participant harbored protease inhibitor (PI) resistance mutations. The rates of drug resistance mutations did not differ significantly between the participants genotyped from DPS (5/43 successfully genotyped samples harbored resistance) and DBS (3/14 successfully genotyped samples harbored resistance) (Fisher’s exact test, *p* = 0.4).

The mutation profiles observed in the eight PDR cases are listed in Table 3. The most commonly observed NNRTI resistance mutations were G190A and Y181C, each observed three times; K103N, K103S, and Y188L were also observed once each. NRTI resistance mutations observed included M184I, M184V, L210W, T215Y, and K219N. In general, children with PDR harbored a single mutation conferring resistance to one or more drugs in the class. Exceptions were EPDOS_8, who harbored NNRTI resistance with K103S and G190A, and EPDOS_53, who harbored dual-class (NNRTI/NRTI) resistance with three NRTI resistance mutations (M184V, L210W, T215Y).

### 3.3. Correlates of PDR in Ethiopian Children

Even though the number of children with observed PDR was small, thereby potentially limiting the power to detect associations, we nevertheless wished to identify correlates of pretreatment HIV drug resistance mutations (Table 4). For this analysis, children were classified as having any (versus no) PDR; however, as all children with PDR harbored NNRTI resistance mutations, this can also be considered an analysis of correlates of NNRTI resistance mutations. Children with PDR were marginally, though not statistically significantly, younger at enrollment [median age 5 (IQR 0.3–10) years among participants with PDR versus 8 (5–12) years in those without PDR; *p* = 0.06]. Moreover, children with PDR had modestly lower albumin levels at HIV diagnosis as compared to their counterparts [2.9 (IQR: 2.5–3.4) mg/dL among participants with PDR versus 3.8 (3.2–4.2) mg/dL among participants with PDR; *p* = 0.04]. Even though a low albumin might suggest undernutrition in HIV-infected children who harbor HIV drug resistance mutations, anthropometric indicators did not show statistically significant differences between children with pretreatment HIV drug resistance mutations and those without. No significant associations were observed between other laboratory and clinical parameters and the presence of HIV drug resistance mutations. There were insufficient cases of NRTI resistance (*N* = 3, all of whom harbored both dual class resistance) to robustly evaluate this category separately; however, we observed that children with dual-class NNRTI/NRTI PDR tended to be younger than those who harbored single or no PDR 0.2 [(0.1–0.3) years for dual-class versus 8 (5–12) for single or none] (*p* = 0.02).

## 4. Discussion

Our study represents the first characterization of pretreatment HIV drug resistance (PDR) among children newly diagnosed with HIV in Ethiopia in 2017–2019. We found that 14% of participants for whom HIV PR-RT sequences could be obtained harbored PDR; of these, approximately two-thirds solely harbored (NNRTI) resistance mutations, and the remainder harbored dual-class (NRTI and NNRTI) resistance mutations. Importantly, all participants with NNRTI resistance had intermediate- to high-level resistance to efavirenz (EFV) and nevirapine (NVP), the NNRTI components of first line cART in Ethiopia. Resistance to PIs was not observed. One important limitation of this study is that we had no information on the PMTCT exposure of the participants. Noting that limitation, the level of HIV drug resistance observed in the present study is comparable to PMTCT-unexposed HIV infected children [39] and is threefold lower than among PMTCT-exposed treatment naïve HIV infected children [39,40] in sub Saharan Africa. A recent analysis of multiple African countries reported a high prevalence of PDR to any antiretroviral drug among infants newly diagnosed with HIV (54.1% overall; 53.0% for NNRTIs and 8.8% for NRTIs) [41]. The much higher prevalence of pediatric PDR in the latter study may be attributable to greater PMTCT exposure, which was reported to be as high as 75–85% in surveys recorded from 2011–2014. 

Our findings also indicate that, in Ethiopia, the burden of PDR among children newly diagnosed with HIV is substantially higher than that in adults newly diagnosed with HIV, in which PDR prevalence has been estimated at 3.9% [42]. Despite the limited PDR data in Ethiopia for both of these patient populations, our observations are consistent with PDR prevalence among adults and children in other high prevalence settings [43]. The difference in the burden of PDR between adults and children could be explained by PMTCT exposure in these children [44], although the observation that the majority of the children studied were born prior to the 2013 implementation of Option B+ suggests only limited PMTCT exposure in this cohort. 

Over the past two decades, rates of HIV PDR in many sub Saharan African regions have been increasing, in some cases to alarming levels [39,45]. Our observations further underscore HIV drug resistance as a major threat to HIV control in resource-limited settings. HIV infected children in Ethiopia are potentially at risk of poor treatment outcomes as a result of high HIV PDR levels [46]. 

Our findings may also have implications for future treatment practices. Currently, the WHO guidelines recommend against the use of an NNRTI-based regimen as a first line treatment if the prevalence of NNRTI PDR in the region exceeds 10% [47]. In Ethiopia, based on the WHO guidelines, PI-based regimens are recommended for children under three years of age, while for children older than three years, two NRTIs (lamivudine with either abacavir, tenofovir, or AZT) plus one NNRTI (either EFV or NVP) are recommended [48,49]. For children older than 10 years, a dolutegravir (DTG) based regimen is recommended [50]. Our observations that 14% of HIV-positive children evaluated had evidence of NNRTI PDR while no children harbored PI resistance supports the consideration of non-NNRTI-based firstline regimens for children of all ages in Ethiopia. Specifically, our findings may support the use of PI-based or DTG-based first regimens in children of all ages. It is acknowledged, however, that the bitter taste of certain pediatric PI-based regimens can be unpalatable, particularly for children, and therefore our findings underscore the urgent need to expand access to newer antiretrovirals and additional drug classes, particularly integrase inhibitors, in Ethiopia.

Some limitations of our study merit mention. The lack of PMTCT information in the EPDOS cohort precludes us from interpreting results in the context of prior antiretroviral exposure and complicates comparisons with other studies from the region. However, as the majority of the children studied were diagnosed at a relatively late age (median nine years), it is possible that these children were only tested subsequent to one or both parents’ recent HIV diagnosis—a common clinical occurrence in Ethiopia. Moreover, most of these children were born before the 2013 scale up of PMTCT Option B+. Taken together, it is likely that most children were not exposed to PMTCT. However, it is important to note that, for those children with prior PMTCT exposure, later diagnosis and enrollment into EPDOS may have allowed resistance mutations associated with NVP exposure to revert to wild-type, leading to an underestimation of the burden of PDR in this cohort. Moreover, our use of Sanger sequencing could have limited our ability to detect low frequency mutations.

In conclusion, the overall prevalence of PDR reported in the current study (14%) is comparable to the prevalence of PDR among PMTCT-unexposed HIV infected children in sub Saharan Africa [39]. As the study sites enrolled HIV infected children with different ethnic and sociodemographic characteristics across two large administrative regions of central and southern Ethiopia, the findings reflect the burden of PDR in Ethiopia at large. The observation that all PDR cases featured mutations that would confer intermediate-to-high-level resistance to efavirenz or nevirapine, the NNRTIs currently available for pediatric first-line treatment in Ethiopia [48,49,50], supports the use of non-NNRTI-based first-line regimens for newly diagnosed HIV infected children in southern Ethiopia (including integrase inhibitor based regimens for eligible children) and calls for establishing a routine drug resistance surveillance in the setting. The relatively low prevalence of NRTI resistance (~5%) and complete lack of PI resistance supports the preferred use of these agents in firstline cART regimens for HIV infected children. Nevertheless, the observation of dual-class PDR, albeit in a minority (5%) of cases, also underscores the urgent need for expanded and affordable access to newer antiretrovirals and additional drug classes, particularly integrase inhibitors, in resource-limited settings.

## Figures and Tables

**Figure 1 viruses-11-00877-f001:**
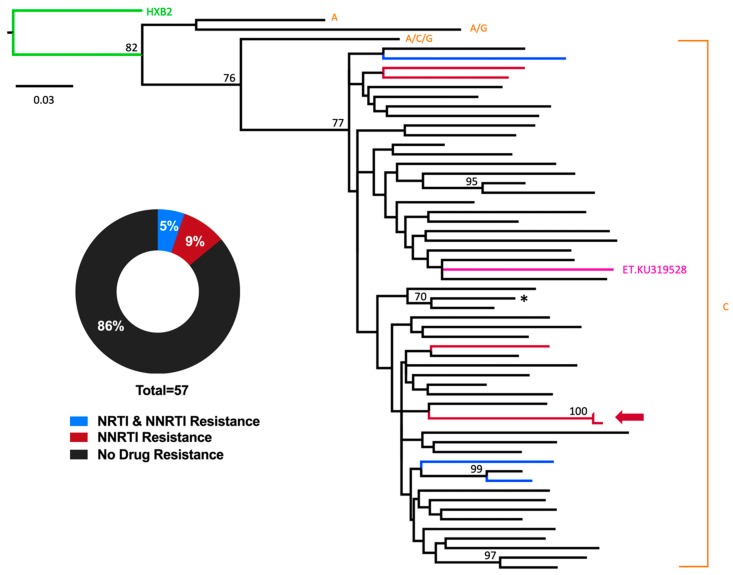
Prevalence of HIV-1 pretreatment HIV drug resistance (PDR) among combination antiretroviral therapies (cART)-naïve Ethiopian children. A maximum-likelihood phylogeny was inferred from the inclusive HIV consensus sequences of the 57 participants for whom genotyping was successful. Drug resistance codons were removed from the alignment prior to phylogenetic inference. Scale indicates expected substitutions per nucleotide site. Nodes with bootstrap values ≥ 70% are indicated on the tree. Colors indicate resistance genotype. HIV-1 subtypes are indicated at tree tips. Reference strains HXB2 (subtype B, green) and KU319528 (subtype C-Ethiopia, pink) are included. The arrow denotes a sibling pair harboring similar HIV sequences. The asterisk denotes a single participant harboring an E138A mutation in reverse transcriptase; this mutation is not on the list of WHO surveillance drug resistance mutations, and therefore this participant is classified in the “no drug resistance” category. However, this mutation confers low-level resistance to the non-nucleoside reverse transcriptase inhibitors (NNRTI) rilpivirine [36].

**Table 1 viruses-11-00877-t001:** Primers used for human immunodeficiency virus (HIV)-1 protease and reverse transcriptase (RT) amplification.

Primer Set		First Round		Second Round	
		HXB2 Coordinates (Start/End)	Sequence (5′→3′)	HXB2 Coordinates (Start/End)	Sequence (5′→3′)
1 *	F	1979/2005	AAGAAGGGCACMTAGCCARAAAYTGYA	2011/2039	CCTAGGAAAAARGGCTGTTGGAARTGTGG
	R	3333/3301	CCACTAACTTCTGTATGTCATTGACAGTCCAGC	3280/3255	ATAGGCTGTACTGTCCATTTATCAGG
2	F	2008/2031	GCCCCTAGGAAAAAGGGCTGTTGG	2011/2039	CCTAGGAAAAARGGCTGTTGGAARTGTGG
	R	3361/3342	TAAATCTGACTTGCCCART	3323/3303	CTGTATRTCATTRACWGTCCA
3	F	1979/2005	AAGAAGGGCACMTAGCCARAAAYTGYA	2011/2039	CCTAGGAAAAARGGCTGTTGGAARTGTGG
	R	3859/3831	GCTCCTACTATGGGTTCTTTYTCYARYTG	3798/3777	CAAACTCCCAYTCAGGRATCCA
4	F	1992/2015	AGCCAGAAATTGCAGGGCCCCTAG	2074/2095	AGACAGGCTAATTTTTTAGGGA
	R	3322/3303	TGTATRTCATTGACAGTCCA	3271/3252	ACTGTCCATTTRTCAGGATG

* indicates the primary primer set. HXB2 is a HIV-1 subtype B reference strain.

**Table 2 viruses-11-00877-t002:** Characteristics of children included in the study.

Variable	Summary Statistic	Total N
Age in years, median (IQR)	9.0 (5.0–12.0)	93
Male, N (%)	48 (51.6)	93
Symptoms at diagnosis, Yes N (%)	56 (65.9)	85
CD4 count, median (IQR) cells/mm^3^	319 (141–615)	41
Plasma viral load in log_10_ copies/mL of, median (IQR)	4.3 (3.7–4.9)	83
Weight for age Z-score, Median (IQR), Z-score	−1.2 (−2.7–(−0.7))	53 *
Height for age Z-score, Median (IQR)	−1.6 (−2.6–(−0.7))	82
Body mass index Z-score, Median (IQR)	−1.1 (−2.2–(−0.1))	82

IQR—interquartile range; normal weight, height, and body mass index Z scores range from −2 to +2. Z-scores between −2 and −3 indicate moderate undernutrition while Z-scores below −3 indicate severe malnutrition. * Weight-for-age Z score was not calculated for some children as the age and/or weight were out of range while using the World Health Organization (WHO) Anthro or AnthroPlus software (https://www.who.int/childgrowth/software/en/).

**Table 3 viruses-11-00877-t003:** Mutational profiles of participants harboring HIV-1 antiretroviral drug resistance.

Sample ID	Dried Spot Type	NRTI Mutations	NNRTI Mutations
EPDOS_6	Plasma	None	G190G/A
EPDOS_8	Plasma	None	K103S, G190A
EPDOS_9	Plasma	None	K103N
EPDOS_22	Plasma	None	G190G/A
EPDOS_29	Plasma	None	Y181C
EPDOS_37	Blood	K219N	Y181C
EPDOS_39	Blood	M184I	Y188L
EPDOS_53	Blood	M184V, L210W, T215Y	Y181C

EPDOS—Efavirenz Pediatric Dose Optimization Study; NRTI—nucleoside reverse transcriptase inhibitors.

**Table 4 viruses-11-00877-t004:** Factors associated with PDR among cART-naïve HIV infected children, Ethiopia, 2018–2019.

Variable	Number Missing	Any Resistance		*p* Value
		Yes (*N* = 8)	No (*N* = 49)	
Age in years, median (IQR)	0	5 (0.3–10)	8 (5–12)	0.06
Sex (% Male)	0	4 (57.1)	24 (51.1)	0.54
WAZ, median (IQR)	23	−1.6 (−2.8–(−0.9))	−1.9 (−2.9–(−0.9))	0.91
HAZ, median (IQR)	7	−1.8 (−1.9–(−0.7))	−1.6 (−3.0–(−0.7))	0.62
BAZ, median (IQR)	7	−1.4 (−2.2–(−1.3))	−1.2 (−2.2–(−0.5))	0.41
CD4, median (IQR), cells/mm^3^	33	267 (217–317)	370 (161–788)	0.53
Log10 pVL, median (IQR) copies/mL	6	4.3 (4.1–4.8)	4.2 (3.8–5.0)	0.47
WHO clinical stage, N (%) 1 2 3 4	4	1 (25) 2 (25.0) 4 (50) 0 (0)	15 (30.6) 10 (20.4) 19 (38.8) 2 (4.1)	0.71

IQR—interquartile range; ALT—alanine aminotransferase; AST—aspartate aminotransferase; HCT—hematocrit; WAZ—weight-for-age Z score; HAZ—height-for-age Z score; BAZ—body mass index-for-age Z score; WHO—World Health Organization.

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
