# Peer review of "Prevalence and Correlates of Pre-Treatment HIV Drug Resistance among HIV-Infected Children in Ethiopia"

_viruses, 2019, doi:10.3390/v11090877_

Round 1

Reviewer 1 Report

The present study investigated pretreatment HIV drug resistance (PDR) burden among children newly diagnosed with HIV in Ethiopia. The authors found that 14% of participants harbored PDR: of these, approximately two-thirds solely harbored NNRTI resistance mutations and the remainder harbored dual-class (NRTI and NNRTI) resistance mutations. Furthermore, all participants with NNRTI resistance had Intermediate to High-Level resistance to efavirenz and nevirapine, the NNRTI components of first line cART in Ethiopia, but no resistance to PIs was observed. This study is important in guiding clinical treatment for HIV infected children in Ethiopia. The authors need to address the following comments. 1) The statement “PDR was marginally associated with decreased baseline serum albumin levels” does not have strong evidence or clinical correlates in PDR patients. Therefore, this statement should be removed from the abstract. 2) The abbreviation (e.g. PT, NRTI, and NNRTI) should be given when used for the first time. 3) Page 4, Line 171: “enrolment” should be “enrollment”. 4) Page 7, lines 247-248, there are two “significantly”, and “enrolment” should be “enrollment”.

Author Response

Response to Reviewer 1 Comments

Reviewer comments are in black, our responses are in red, with text quoted in the revised manuscript in italics.

Comment 1: The present study investigated pretreatment HIV drug resistance (PDR) burden among children newly diagnosed with HIV in Ethiopia. The authors found that 14% of participants harbored PDR: of these, approximately two-thirds solely harbored NNRTI resistance mutations and the remainder harbored dual-class (NRTI and NNRTI) resistance mutations. Furthermore, all participants with NNRTI resistance had Intermediate to High-Level resistance to efavirenz and nevirapine, the NNRTI components of first line cART in Ethiopia, but no resistance to PIs was observed. This study is important in guiding clinical treatment for HIV infected children in Ethiopia. The authors need to address the following comments. 1) The statement “PDR was marginally associated with decreased baseline serum albumin levels” does not have strong evidence or clinical correlates in PDR patients. Therefore, this statement should be removed from the abstract. 2) The abbreviation (e.g. PT, NRTI, and NNRTI) should be given when used for the first time. 3) Page 4, Line 171: “enrolment” should be “enrollment”. 4) Page 7, lines 247-248, there are two “significantly”, and “enrolment” should be “enrollment”.

Response 1: Thank you for the positive comments on our manuscript. We have removed the statement linking albumin levels to PDR from the abstract and we have corrected all typographic errors

Reviewer 2 Report

The manuscript by Birkneh Tilahun Tadesse et al is a well-written and well-executed research studying the prevalence of transmitted drug-resistance mutations in HIV-infected children from two large regions in Ethiopia. Overall, I have only one major concern, namely that the numbers of PDR found are limited, so that the statistical analyses are not as robust as the authors would like. I suggest adding some caution to the text where appropriate, such as in sections 3.2 and 3.3 (on the correlates of PDR), especially when splitting the low numbers into even smaller categories.  

Minor comments:

- lines 68-69: it is stated her that ‘the majority of children in Ethiopia has been exposed to some sort of PMTCT intervention’. Could the authors explain in more detail? Are all pregnant women treated without testing them for HIV? What is the prevalence of HIV-infection in the adult population in Ethiopia?

- line 173: the mean age of the study participants was 9 years (range 5-12), implying that they were infected with HIV approx. 9 years ago. Did PMTCT policy change over time? Would you still expect similar levels in children born in 2018/2019?

- line 275: please change ‘newly’ in ‘diagnosed in 2018-2019’ for future readers.

- lines 297-298: What ‘pediatric populations’ have been identified here ‘as being particularly at risk’? Please specify, as the eight cases appear to be quite random.

- lines 320-324: the authors propose here to use PI-based regimens as first-line treatment for HIV-infected children in Ethiopia and advocate affordable access to newer antivirals. Would that be wise, as it will probably not decrease drug-resistance problems without implementation of routine virologic testing and sequencing? In addition, depending on the genetic make-up, the co-formulated PI’s lopinavir/ritonavir (Kaletra), the backbone of the current first-line antiretroviral regimens for young children, can taste very bitter and is refused by many. Please comment.

Author Response

Response to Reviewer 2 Comments

Reviewer comments are in black, our responses are in red, with text quoted in the revised manuscript in italics.

Comment 1: The manuscript by Birkneh Tilahun Tadesse et al is a well-written and well-executed research studying the prevalence of transmitted drug-resistance mutations in HIV-infected children from two large regions in Ethiopia. Overall, I have only one major concern, namely that the numbers of PDR found are limited, so that the statistical analyses are not as robust as the authors would like. I suggest adding some caution to the text where appropriate, such as in sections 3.2 and 3.3 (on the correlates of PDR), especially when splitting the low numbers into even smaller categories.  

Response 1: Thank you for the comments on the quality of our study and our paper. We have added a line in the results to address the reviewer's concerns regarding limited statistical power due to relatively limited PDR cases, as follows (Section 3.3, Paragraph 1) to address the limitation. 

Even though the number of children with observed PDR was small, thereby potentially limiting the power to detect associations, we nevertheless wished to identify correlates of pretreatment HIV drug resistance mutations (Table 4)

Minor comments:

Comment 2: lines 68-69: it is stated her that ‘the majority of children in Ethiopia has been exposed to some sort of PMTCT intervention’. Could the authors explain in more detail? Are all pregnant women treated without testing them for HIV? What is the prevalence of HIV-infection in the adult population in Ethiopia?

Response 2: We have removed the confusing statements regarding child exposure to PMTCT from the introduction and instead now elaborate on this issue in the methods (see response to comment 4 below). We have also added a line in the introduction that reports HIV prevalence in the adult population in Ethiopia as 1.4% overall.

Comment 3: line 173: the mean age of the study participants was 9 years (range 5-12), implying that they were infected with HIV approx. 9 years ago. Did PMTCT policy change over time? Would you still expect similar levels in children born in 2018/2019?

Response 3: As outlined in the introduction, PMTCT coverage in Ethiopia has been improving steadily since the policy change in 2013 which required initiation of cART for all HIV infected pregnant women. The PMTCT exposure in 2018/2019 is therefore higher than a decade back. We now provide additional information on PMTCT availability in Ethiopia is now provided on Section 2.1, Paragraph 2.

Beginning in 2001, Ethiopia adopted PMTCT intervention Option A, under which eligible pregnant women with CD4 <350 copies/mm3 were initiated on cART. At this time, women who did not meet the CD4-based eligibility criteria were provided antepartum AZT and intrapartum single dose nevirapine. In 2013, the guidelines were amended to recommend Option B+ and uptake has scaled up since then. By 2014, around 2500 health facilities had started providing PMTCT services. Currently, PMTCT coverage estimates in Ethiopia vary between 50-70% depending on region [11, 16, 17].

Comment 4: line 275: please change ‘newly’ in ‘diagnosed in 2018-2019’ for future readers.

Response 4: We have revised this sentence as follows:

Our study represents the first characterization of pretreatment HIV drug resistance (PDR) among children newly diagnosed with HIV in Ethiopia in 2017-19.

Comment 5:  lines 297-298: What ‘pediatric populations’ have been identified here ‘as being particularly at risk’? Please specify, as the eight cases appear to be quite random.

Response 5: Apologies for the confusion.  We were referring to HIV infected children in general and have revised the sentence as follows.

Our observations further underscore HIV drug resistance as a major threat to HIV control in resource-limited settings. HIV infected children in Ethiopia are potentially at risk of poor treatment outcomes as a result of high HIV PDR levels [46].

Comment 6: lines 320-324: the authors propose here to use PI-based regimens as first-line treatment for HIV-infected children in Ethiopia and advocate affordable access to newer antivirals. Would that be wise, as it will probably not decrease drug-resistance problems without implementation of routine virologic testing and sequencing? In addition, depending on the genetic make-up, the co-formulated PI’s lopinavir/ritonavir (Kaletra), the backbone of the current first-line antiretroviral regimens for young children, can taste very bitter and is refused by many. Please comment.

Response 6: We appreciate this comment. In the current national guidelines, PI based regimens are recommended as first line regimens for children <3 years and as second line regimens for those >3 years of age. Despite the difficult taste especially for younger children, a PI-based regimen is currently the only option for younger children who have been exposed to NNRTIs through PMTCT.  We have revised this section of the manuscript as follows (Section 4, Paragraph 4):

Our findings may also have implications for future treatment practises. Currently, the WHO guidelines recommend against the use of an NNRTI-based regimen as a first line treatment if the prevalence of NNRTI PDR in the region exceeds 10% [47]. In Ethiopia, based on the WHO guidelines, PI-based regimens are recommended for children under 3 years of age, while for children older than three years, recommended first line regimens include two NRTIs (3TC with either ABC, TDF, or AZT) plus one NNRTI (either EFV or NVP) and second line regimens consist of two NRTIs (3TC and additional NRTI not included in the initial regimen) and a Protease Inhibitor (PI) [48, 49]. For children older than 10 years, a dolutegravir (DTG) based regimen is recommended [50]. Our observations that 14% of HIV-positive children evaluated had evidence of NNRTI PDR while no children harbored PI resistance supports the consideration of non-NNRTI-based firstline regimens for children of all ages in Ethiopia. Specifically, our findings may support the use of PI-based or DTG-based regimens in children of all ages. It is acknowledged however that the bitter taste of certain pediatric PI-based regimens can be unpalatable, particularly for children, and therefore our findings underscore the urgent need to expand access to newer antiretrovirals and additional drug classes, particularly integrase inhibitors, in Ethiopia.

We agree with the comment on routine drug resistance testing and we have included a section in the discussion to address that. Please see section 4 (Conclusions).

… and calls for establishing a routine drug resistance surveillance in the setting.

Reviewer 3 Report

This manuscript reports on levels of pre-treatment HIV drug resistance (PDR) in ART-naïve children aged 3-12 years in two regions of Ethiopia. Dried blood spots (DBS) or dried plasma spots (DPS) were collected from children at enrolment in a prospective cohort study (the Efavirenz Pediatric Dose Optimization Study). Sanger sequencing was performed and drug resistance was interpreted using the Calibrated Population Resistance tool, based on the WHO 2009 list of surveillance drug resistance mutations (SDRM).

Of 111 enrolled in the cohort, a valid pol sequence was only available for 57 – there was attrition from lack of a pre-ART specimen (n=18) and unsuccessful amplification/sequencing from the DBS/DPS (n=36). The overall proportion with any SDRM was 14% - 5 participants had only NNRTI resistance and 3 participants had dual class (NRTI/NNRTI) resistance.

The paper is generally well written and the descriptive analysis of pre-treatment drug resistance is appropriate. It is a useful contribution to the HIVDR literature, as there is a relative paucity of data in children of this age and of HIVDR data from that part of Africa.

Major comments

The analysis exploring ‘correlates’ of PDR does not seem to have been guided by a clear question. Rather it seems to just explore the association between PDR and a number of variables (demographic, clinical, laboratory markers) which were available as part of the study. As a result, the finding that PDR is associated with lower albumin is most likely to be a chance finding. I would recommend revisiting this part of the analysis with a clear question guiding the analysis – perhaps focusing on the variables where there might be a biologically plausible association with PDR (particularly age, sex, viral load). The authors could also consider then building a logistic regression model to explore the association with these key variables.

The lack of data on maternal or child exposure to antiretrovirals (for pMTCT) is a major limitation in interpreting the findings. Did the authors really have no data at all or were the data incomplete? On p2, line 95 it is stated that ‘…PMTCT history was not comprehensively available for study participants’ whereas on p8, line 280-281 it is stated that ‘…we had no information on the PMTCT exposure of the participants.  This is a serious limitation of the study – if the authors have at least some data then it would be helpful to interpret the results.

Minor comments

There is confusion in HIV drug resistance terminology. Pre-treatment drug resistance (PDR) is not the same as primary or transmitted drug resistance (it includes resistance from prior exposure to antiretrovirals – for example, as in this case, from prior pMTCT exposure). The authors should stick with pre-treatment drug resistance throughout to avoid confusion.

Methods: Given the lack of data on pMTCT exposure, there should at least be clear details of the dates of enrolment and a more detailed description of the evolution of pMTCT guidelines in Ethiopia (and ideally levels of pMTCT coverage) so that the reader can get some sense of the likelihood that there was maternal/child exposure to ARVs. Although in the introduction the authors state that Ethiopia moved to option B+ in 2013, it seems that most of the children in the study would have been born before that time.

Methods, p2, line 90: please clarify age inclusion criteria – here it states 3-12 years, whereas in paper describing the cohort (ref. 13) it states ages were 3-18 years.

Methods, p2, line 91: suggest being explicit about how ART-naïve status was determined – was this just self-report of parent/caregiver or did the investigators have a way of checking medical record systems/HIV programme databases?

Methods, p2, lines 91 & 93: Presumably the children with acute severe illnesses and tuberculosis were excluded for reasons related to the parent study, not to this specific drug resistance substudy? It might be worth making this clear.

Table 1: The table with the simple description of primers is not necessary as this is not a laboratory methods paper.

Results, p4, lines 173-176: These two sentences and how staging changed after ‘further clinical evaluation’ are not needed. Suggest just describing whichever was the better reflection of true clinical stage.

Results, p4, line 177: typo - popular should be papular

Results, p5, lines 179-182: the Karnofsky score is not meant to be applied to children. This can be left out anyway as it has no direct relevance to your focus on drug resistance.

Results, p5: general point – rather than Q1-Q3, should state interquartile range (IQR)

Results, p5, lines 199-203: It is surprising that success of genotyping was not associated with viral load. It would be worth stating this explicitly, if it is indeed so. Again the association with weight-for-age score is hard to explain and most likely a chance finding – The authors should carefully contextualise this finding and explain its relevance.

Results, p5, line 206-207: This suggests that the authors did not exclude poor quality sequences. Is this correct? Usually incomplete or poor quality sequences would be excluded from analysis. Did the authors include incomplete / poor quality sequences as it seems that they still had coverage of all SDRM positions?

Results, p6, line 227: It would be helpful to give 95% confidence intervals for the proportion here, as it will allow the reader to see the uncertainty around the estimate.

Discussion: The discussion is useful of how the findings relate to other studies of PDR in children, which have most commonly been done with infants and younger children. There should, however, be more attention paid to the fact that the study population was older children it is possible that transmitted mutations (or mutations acquired from NVP use) wane over time. So it’s quite possible that the findings underestimate the true level of resistance. In that light the authors should also discuss that the only did Sanger sequencing. Next-generation sequencing would have improved their ability to detect low-frequency mutations (minority variants).

The discussion centres on the need to move to PI-based first-line regimens – it is not clear if the authors are arguing this for children of all ages? The common PI (lopinavir/ritonavir) is very poorly tolerated in children and outcomes can be poor. Do the authors really think that is the appropriate public health intervention? The global trend is a move towards integrase inhibitors – there should be more discussion of whether there are plans in Ethiopia to transition to dolutegravir-based regimens for children and whether these findings should influence policy makers in this regard.

Author Response

Response to Reviewer 3 Comments

Reviewer comments are in black, our responses are in red, with text quoted in the revised manuscript in italics.

Major comments

Comment 1: The analysis exploring ‘correlates’ of PDR does not seem to have been guided by a clear question. Rather it seems to just explore the association between PDR and a number of variables (demographic, clinical, laboratory markers) which were available as part of the study. As a result, the finding that PDR is associated with lower albumin is most likely to be a chance finding. I would recommend revisiting this part of the analysis with a clear question guiding the analysis – perhaps focusing on the variables where there might be a biologically plausible association with PDR (particularly age, sex, viral load). The authors could also consider then building a logistic regression model to explore the association with these key variables.

Response 1: Thank you for the comment. We have removed all variables from the analysis where there is no strong rationale for finding a biological association with PDR in a revised Table 4. As none of the remaining variables were statistically significantly associated with PDR in the univariable analysis, we did not explore multivariable logistic regression models.

Comment 2: The lack of data on maternal or child exposure to antiretrovirals (for pMTCT) is a major limitation in interpreting the findings. Did the authors really have no data at all or were the data incomplete? On p2, line 95 it is stated that ‘…PMTCT history was not comprehensively available for study participants’ whereas on p8, line 280-281 it is stated that ‘…we had no information on the PMTCT exposure of the participants.  This is a serious limitation of the study – if the authors have at least some data then it would be helpful to interpret the results.

Response 2: Unfortunately, PMTCT information was not available in the parent study (EPDOS). We acknowledge that the absence of these data are a limitation of our study. However, as most of these children were first diagnosed with HIV at a relatively late age (median age of 9 year), it is plausible that these children were only tested subsequent to one or both parents' recent HIV diagnosis. This type of clinical scenario occurrs commonly in our setting. In these cases, it is likely that the children had not been exposed to PMTCT. We have added the following section to the Discussion section (Section 4, Paragraph 5):

Some limitations of our study merit mention. The lack of PMTCT information in the EPDOS cohort precludes us from interpreting results in the context of prior antiretroviral exposure and it complicates comparisons with other studies from the region. However, as the majority of the children studied were diagnosed at a relatively late age (median 9 years), it is possible that these children were only tested subsequent to one or both parents' recent HIV diagnosis - a common clinical occurrence in Ethiopia. Moreover, most of these children were born before the 2013 scale up of PMTCT Option B+. Taken together, it is likely that most children were not exposed to PMTCT. However, it is important to note that for those children with prior PMTCT exposure, later diagnosis and enrollment into EPDOS may have allowed resistance mutations associated with NVP exposure to revert to wild-type leading to an underestimation the burden of PDR in this cohort. Moreover, our use of Sanger sequencing could have limited our ability to detect low frequency mutations.

Minor comments

Comment 3: There is confusion in HIV drug resistance terminology. Pre-treatment drug resistance (PDR) is not the same as primary or transmitted drug resistance (it includes resistance from prior exposure to antiretrovirals – for example, as in this case, from prior pMTCT exposure). The authors should stick with pre-treatment drug resistance throughout to avoid confusion.

Response 3: We now refer to “pre-treatment drug resistance (PDR)” throughout the revised version.

Comment 4: Methods: Given the lack of data on pMTCT exposure, there should at least be clear details of the dates of enrolment and a more detailed description of the evolution of pMTCT guidelines in Ethiopia (and ideally levels of pMTCT coverage) so that the reader can get some sense of the likelihood that there was maternal/child exposure to ARVs. Although in the introduction the authors state that Ethiopia moved to option B+ in 2013, it seems that most of the children in the study would have been born before that time.

Response 4: Thank-you for this comment. We have included information regarding PMTCT interventions in the methods section under 2.1 as well as a discussion of this issue in the limitations section. (please see response to comments 4 and 9, above).

Comment 5: Methods, p2, line 90: please clarify age inclusion criteria – here it states 3-12 years, whereas in paper describing the cohort (ref. 13) it states ages were 3-18 years.  

Response 5: Thank you - this was a typo and has been corrected. The enrollment criteria were identical to the parent study i.e. 3-18 years.

Comment 6: Methods, p2, line 91: suggest being explicit about how ART-naïve status was determined – was this just self-report of parent/caregiver or did the investigators have a way of checking medical record systems/HIV programme databases?

Response 6: Children were included in the study when they were diagnosed as HIV-positive for the first time either during testing in acute care settings or during community testing programs. As the HIV care and treatment programs in Ethiopia are managed using a centralized database, we could confirm whether a child was previously enrolled in care. As such, we are confident that these children were cART naïve at the time of enrollment into the EPDOS cohort. We have now clarified this in the methods section 2.1.

Comment 7: Methods, p2, lines 91 & 93: Presumably the children with acute severe illnesses and tuberculosis were excluded for reasons related to the parent study, not to this specific drug resistance substudy? It might be worth making this clear.  

Response 7: That is correct. We have clarified that in the methods section 2.1.

Children who had tuberculosis and those who had previously been on combination antiretroviral therapy were not eligible to be enrolled in the parent EPDOS study, though children with PMTCT exposure would have been eligible if data were available. Note however that PMTCT history was not available for study participants.

Comment 8: Table 1: The table with the simple description of primers is not necessary as this is not a laboratory methods paper.

Response 8: We appreciate this comment however we respectfully wish to keep this information. Even though this is not a laboratory methods paper it may nevertheless be helpful for laboratories to have access to the primers used to successfully amplify HIV subtype C sequence from our region.

Comment 9: Results, p4, lines 173-176: These two sentences and how staging changed after ‘further clinical evaluation’ are not needed. Suggest just describing whichever was the better reflection of true clinical stage. Results, p4, line 177: typo - popular should be popular. Results, p5, lines 179-182: the Karnofsky score is not meant to be applied to children. This can be left out anyway as it has no direct relevance to your focus on drug resistance.  Results, p5: general point – rather than Q1-Q3, should state interquartile range (IQR)

Response 9: Clinical staging was removed but symptoms description retained. We have also removed the text related to the Karnofsky scoring. Corrected the typo errors, changed Q1-Q3 to IQR. This section now reads:

Upon clinical evaluation, 56/85 (65.9%) were identified as having symptoms that define WHO clinical stage 2 or above (Table 1). These symptoms included papular pruritic eruptions (21/85; 24.7%), mucocutaneous viral infections (24/85; 28.2%), chronic diarrhea (18/83; 21.2%) and features of fungal infection (15/85; 17.7%).

Comment 10: Results, p5, lines 199-203: It is surprising that success of genotyping was not associated with viral load. It would be worth stating this explicitly, if it is indeed so. Again the association with weight-for-age score is hard to explain and most likely a chance finding – The authors should carefully contextualise this finding and explain its relevance.

Response 10: We acknowledge that this is surprising, and we now explicitly stated this in the text on Section 3.2, Paragraph 1. 

Somewhat surprisingly, samples for which resistance genotyping was successful did not have significantly different viral loads from those samples were resistance was not successful (Median 4.3 [IQR: 3.8-5.0] vs 4.4 [IQR: 3.2-4.9] p=0.47]

We also acknowledge later in this same paragraph that the relationship between nutritional status and success may be a chance finding.

Comment 11: Results, p5, line 206-207: This suggests that the authors did not exclude poor quality sequences. Is this correct? Usually incomplete or poor quality sequences would be excluded from analysis. Did the authors include incomplete / poor quality sequences as it seems that they still had coverage of all SDRM positions?

Response 11: We apologize for any confusion. Poor quality or incomplete sequences were considered “unsuccessful” and were excluded from the analysis.  This section  was intended to report that, despite starting from small volume dried blood spots or dried plasma spots, the sequencing protocol was able to identify within-host sequence variation (in the form of ambiguous nucleotide “mixtures”) in a subset of samples.

Comment 12: Results, p6, line 227: It would be helpful to give 95% confidence intervals for the proportion here, as it will allow the reader to see the uncertainty around the estimate.

Response 12: This has been done.

Comment 13: Discussion: The discussion is useful of how the findings relate to other studies of PDR in children, which have most commonly been done with infants and younger children. There should, however, be more attention paid to the fact that the study population was older children it is possible that transmitted mutations (or mutations acquired from NVP use) wane over time. So it’s quite possible that the findings underestimate the true level of resistance.

Response 13: Thank-you for the positive comments on our discussion. Regarding waning of transmitted drug resistance, we have now added the following comment to the study:

....it is important to note that for those children with prior PMTCT exposure, later diagnosis and enrollment into EPDOS may have allowed resistance mutations associated with NVP exposure to revert to wild-type leading to an underestimation the burden of PDR in this cohort.

Comment 14: In that light the authors should also discuss that the only did Sanger sequencing. Next-generation sequencing would have improved their ability to detect low-frequency mutations (minority variants).

Response 14: We acknowledge that NGS may have allowed more sensitive detection of low-abundance mutations; however, we do note that the majority of similar PDR surveys in adult and pediatric populations continue to rely on Sanger sequencing. As such, the use of Sanger sequencing here may allow the results of this study to be put into context with other studies from the region. We nevertheless have added the following statement to the discussion of limitations.

Moreover, our use of Sanger sequencing could have limited our ability to detect low frequency mutations

Comment 15: The discussion centres on the need to move to PI-based first-line regimens – it is not clear if the authors are arguing this for children of all ages? The common PI (lopinavir/ritonavir) is very poorly tolerated in children and outcomes can be poor. Do the authors really think that is the appropriate public health intervention? The global trend is a move towards integrase inhibitors – there should be more discussion of whether there are plans in Ethiopia to transition to dolutegravir-based regimens for children and whether these findings should influence policy makers in this regard.

Response 15: At the present time, Ethiopia is rolling out the scale up DTG for adults and adolescents with some changes related to the recent safety concerns for women of reproductive age. The 2018 Ethiopian guidelines recommend AZT or ABC + 3TC + EFV as first line for children 3-10 years while TDF + 3TC + DTG (FDC) OR TDF + 3TC + EFV (FDC) is recommended for adolescents >10 years. For children <3 years, ABC or AZT + 3TC + LPV/r is the recommended first line cART. We agree with the assertion that PI based regimens are poorly tolerated and could be associated with poorer outcome. The guidelines nevertheess keep LPV/r as a preferred second line regimen for children who failed NNRTI based first line regimens.

As described in our response to comment #4 above, we have now revised the discussion considering the national guidelines for cART in children.

Round 2

Reviewer 3 Report

The authors have addressed my comments